# A Modified Xception Deep Learning Model for Automatic Sorting of Olives Based on Ripening Stages

Seyed Iman Saedi [1] and Mehdi Rezaei [2,*]

1   Department of Water and Soil, Faculty of Agriculture, Shahrood University of Technology, Shahrood 3619995161, Iran; isaedi@shahroodut.ac.ir
2   Department of Horticulture, Faculty of Agriculture, Shahrood University of Technology, Shahrood 3619995161, Iran
*   Correspondence: mhrezaei@shahroodut.ac.ir

**Abstract:** Olive fruits at different ripening stages give rise to various table olive products and oil qualities. Therefore, developing an efficient method for recognizing and sorting olive fruits based on their ripening stages can greatly facilitate post-harvest processing. This study introduces an automatic computer vision system that utilizes deep learning technology to classify the 'Roghani' Iranian olive cultivar into five ripening stages using color images. The developed model employs convolutional neural networks (CNN) and transfer learning based on the Xception architecture and ImageNet weights as the base network. The model was modified by adding some well-known CNN layers to the last layer. To minimize overfitting and enhance model generality, data augmentation techniques were employed. By considering different optimizers and two image sizes, four final candidate models were generated. These models were then compared in terms of loss and accuracy on the test dataset, classification performance (classification report and confusion matrix), and generality. All four candidates exhibited high accuracies ranging from 86.93% to 93.46% and comparable classification performance. In all models, at least one class was recognized with 100% accuracy. However, by taking into account the risk of overfitting in addition to the network stability, two models were discarded. Finally, a model with an image size of 224 × 224 and an SGD optimizer, which had a loss of 1.23 and an accuracy of 86.93%, was selected as the preferred option. The results of this study offer robust tools for automatic olive sorting systems, simplifying the differentiation of olives at various ripening levels for different post-harvest products.

**Keywords:** olive; color image; Xception; sorting; deep learning





## 1. Introduction

Olive, *Olea europaea*, is an essential evergreen subtropical fruit. Its fruits are utilized for both table olives and olive oil. Certain varieties are specifically cultivated for oil production, while others, renowned for their larger fruit sizes, are preferred for canning products. Moreover, the production of dual-purpose olive varieties is growing [1]. In Iran, the 'Roghani' cultivar stands out as a vital local dual-purpose olive variety, known for its adaptability to diverse environmental conditions and ability to withstand winter cold [2]. The type of canned olives and the quality of olive oil depend on various factors, including the variety, cultivation conditions, and fruit ripening stage [3–5]. Olive fruits can be harvested at different stages, ranging from immature green to fully mature black, and even during over-ripened stages. The ripening stage of the fruit profoundly affects the oil content, chemical composition, sensory characteristics of olive oil, and industrial yield [6,7]. Fruit homogeneity at the same ripening stage is crucial for canned olives, and the quality of olive oil directly depends on the fruit's ripening stage.

The timing of olive harvesting is typically determined by evaluating the maturity index (MI) of each olive cultivar [5,8,9]. This evaluation of MI is based on changes in both

the skin and flesh color of mature fruit [10]. Decisions about when to harvest fruit from an orchard are made by conducting MI assessments on fruit samples collected from different trees. However, it is common to come across olives with varying degrees of ripeness during processing, as mechanical harvesters that use trunk shakers can harvest one hectare of an intensive olive grove (consisting of 300–400 trees) within a timeframe of 2 to 5 days [10]. Due to factors such as the location of the fruit on outer or inner branches and exposure to sunlight, even a single tree may have olives in different stages of maturity, and there may be variations between each tree due to differences in horticultural practices and management. Moreover, some orchards may cultivate multiple olive varieties, each with distinct ripening stages during harvest, while others with a single cultivar may also have variations in fruit ripeness. In olive processing facilities, it is possible for different growers to bring olives with varying degrees of ripeness that must be categorized before processing.

Given the importance of olive ripening in the production of various post-harvest products, such as pickles, oil, and canned olives, it is essential to separate and sort olive fruits before processing. However, manually sorting olives through human visual inspection is a challenging and inefficient task. To address this challenge, integrating a computer vision system into olive processing units as part of the automatic separation machinery offers a potential solution. The system consists of an image-capturing unit, which relies on a robust image processing model to ensure rapid and accurate results for mechanical separation [11].

Numerous researchers have investigated various methods for assessing olive fruit maturity, with a focus on Near Infrared Spectroscopy (NIRS) [10,12,13]. These studies aimed to predict diverse quality parameters and characterize table olive traits utilizing NIRS technology. In addition to NIRS, Convolutional Neural Networks (CNNs), a subset of deep learning, have emerged as a powerful tool for image processing tasks, allowing for the extraction of high-level features independent of imaging condition and structure [14], making them a valuable tool for agricultural applications.

The use of cutting-edge technologies, such as deep learning, offers a more promising solution to address this challenge, garnering the attention of scientists across multiple agricultural domains [15–17]. Noteworthy applications of CNNs include olive classification, as demonstrated by Riquelme et al. [18], who employed discriminant analysis to classify olives based on external damage in images, achieving validation accuracies ranging from 38% to 100%. Guzmán et al. [9] leveraged algorithms based on color and edge detection for image segmentation, resulting in an impressive 95% accuracy in predicting olive maturity. Ponce et al. [19] utilized the Inception-ResNetV2 model to classify seven olive fruit varieties, achieving a remarkable maximum accuracy of 95.91%. Aguilera Puerto et al. [20] developed an online system for olive fruit classification in the olive oil production process, employing Artificial Neural Networks (ANN) and Support Vector Machines (SVM) to attain high accuracies of 98.4% and 98.8%, respectively. Aquino et al. [21] created an artificial vision algorithm capable of classifying images taken in the field to identify olives directly from trees, enabling accurate yield predictions. Studies such as Khosravi et al. [17] have also utilized RGB image acquisition and CNNs for the early estimation of olive fruit ripening stages on-branch, which has direct implications for orchard production quality and quantity. Furferi et al. [22] proposed an ANN-based method for automatic maturity index evaluation, considering four classes based on olive skin and pulp color, while ignoring the presence of defects. In contrast, Puerto et al. [23] implemented a static computer vision system for olive classification, employing a shallow learning approach using an ANN with a single hidden layer. In a recent study by Figorilli et al. [24], olive fruits were classified based on the state of external color, "Veraison", and the presence of visible defects using AI algorithms with RGB imaging. Despite the commendable efforts in olive fruit detection modeling, however, previous studies have primarily focused on identifying defective olives, inadvertently overlooking the comprehensive assessment of distinct stages crucial for olive fruit ripening, impacting both oil quality and canned olive production [18,20]. This trend, compounded by the reliance on limited datasets, has significantly hindered the models' capacity for

effective generalization. Moreover, the dependency of these models on specific imaging modalities has constrained their adaptability to varying environmental conditions, lighting disparities, and diverse olive cultivars. Additionally, one of our primary objectives is to enhance model efficiency and recognition speed. Therefore, we endeavored to address these gaps by conducting a comprehensive exploration of olive fruit ripening stages and developing new high-performance models to better address this critical area.

The field of machine learning has seen significant advancements in recent years, particularly in agriculture. According to Benos et al. [25], there was a remarkable 745% increase in articles related to machine learning in agriculture between 2018 and 2020, indicating the growing use of machine learning algorithms for crop and animal analysis based on input data from satellites and drones. This surge in interest is attributed to the development of novel models that exhibit high performance and optimized detection times. For instance, Fan et al. [26], successfully utilized a YOLOV4 network to detect defects in apple fruits using near-infrared (NIR) images, achieving an average detection accuracy of 93.9% and processing five fruits per second.

In the realm of fruit recognition, the Xception deep learning model has been gainfully employed [27]. Built upon the Inception architecture, Xception is a powerful neural network that excels in image classification tasks owing to its efficiency and accuracy [28]. By taking the concept of separable convolutions to an extreme level, Xception becomes a highly efficient and powerful network, demonstrating the potential of CNNs for image processing tasks.

This study aims to leverage the Xception deep learning model for the automated sorting of olives based on color images, given the critical role of olive fruit sorting in producing diverse end products (e.g., pickles, oil, canned goods). Our ultimate goal is to create a highly accurate and robust computer vision system capable of categorizing Roghani olives into five distinct ripening stages. We evaluated the system's performance using test dataset accuracy, classification performance metrics (such as classification reports and confusion matrices), and its capacity to generalize across varied datasets.

The significance of our research lies in its potential to offer olive processing facilities efficient and reliable tools for automating the sorting process, thus distinguishing between olives of differing ripeness levels. This, in turn, may enhance the quality of various post-harvest products and differentiate olive oil qualities, ultimately benefiting the olive industry as a whole. By providing a more accurate method for sorting olives according to their maturity, we can improve the overall quality of downstream products such as pickles, oil, and canned goods. Moreover, our proposed approach could potentially reduce waste and increase efficiency within the olive processing industry.

## 2. Material and Methods

### 2.1. Data Preparation

To develop an image-based CNN model for classifying olive fruits based on their ripening stages, we considered an Iranian olive cultivar named Roghani at five distinct ripening stages. A total of 761 images of different classes were captured in an unstructured laboratory setting using a smartphone camera (Samsung Galaxy A51, India). One of the most important advantages of deep learning-based image processing techniques and convolutional neural networks is that unlike the traditional image processing methods, they are not dependent on environmental conditions including lighting, background, distance to object, camera properties, etc., making them more powerful and robust tools in processing images taken in natural conditions. By addressing this fact, we did not consider a special capturing condition or structure. The captured images had an initial resolution of $3000 \times 4000$ pixels. Figure 1 depicts the five ripening stages of the olive fruits and their corresponding average mass. The color attributes of the samples served as the basis for discriminating between ripening stages. Specifically, Stage 1 refers to samples with green colors, Stage 2 is characterized by olives with 10–30% browning, while Stages 3, 4, and 5 represent approximately 50, 90, and 100% browning (fully black), respectively. The number

of images taken at each ripening stage, and the average mass of samples at each class, are presented in Table 1.

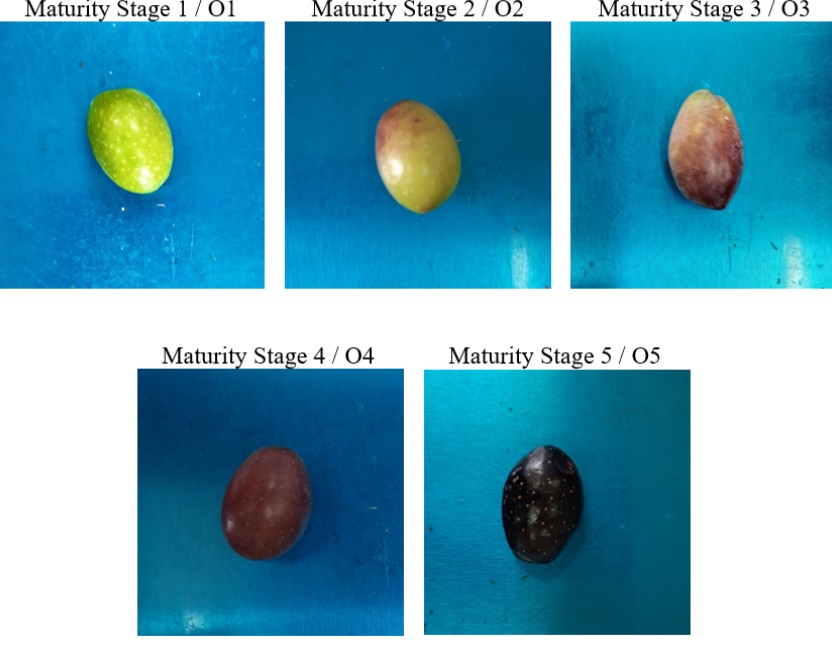

**Figure 1.** Adjusted images of the five ripening stages of Roghani olives under our study. Each class has been denoted by an assigned code (O1–O5).(Original pictures).

**Table 1.** The number of images taken at each stage of olive fruit maturity and the average mass of samples at each class.

| Classes | O1 | O2 | O3 | O4 | O5 |
|---|---|---|---|---|---|
| Number of Samples | 195 | 161 | 183 | 93 | 129 |
| Average Mass (g) | $4.05 \pm 1.30$ | $2.93 \pm 0.90$ | $3.00 \pm 0.78$ | $3.22 \pm 0.67$ | $3.74 \pm 3.29$ |

The image data was divided into three distinct parts: the training set, the validation set, and the testing set. To accomplish this, 20% of the total data (equivalent to 153 images) were assigned to the test dataset. The remaining data consisted of 609 images, with 15% (92 images) being allocated for the validation set, and the remaining 516 images being utilized for the training set.

The training process involved passing the input data through several layers, obtaining the output, and comparing it with the desired output. The difference between the two, which served as the error, was then calculated. Using this error, the network parameters were adjusted and fed the data back into the network to compute new results and errors. This process was repeated multiple times, adjusting the parameters after each iteration to minimize the error. There are various formulas and functions to calculate the network error. Once the error was computed, the parameters were updated to move closer to minimizing it; that is, optimizing the weights to achieve the lowest possible error.

Preprocessing the input images is crucial to enhance the model's accuracy, prevent overfitting, and boost its generalization capability. First, we resized all images to two different sizes: $224 \times 224$ and $299 \times 299$. Next, we normalized the pixel values by dividing them by the maximum pixel values of the captured images. Subsequently, we applied data augmentation techniques, including random translation, random flip, random contrast, and random rotation, to artificially increase the number of images used in model development. The data augmentation parameters are presented in Table 2.

**Table 2.** Augmentation arguments and values.

| Augmentation Parameters | Value |
| --- | --- |
| Random Translation (height_factor) | 0.1 |
| Random Translation (width_factor) | 0.1 |
| Random Flip | True |
| Random Contrast | 0.15 |
| Random Rotation | 0.15 |

To develop the deep neural network model, we utilized the transfer learning technique. Initially, we invoked the Xception model and loaded its weights from the ImageNet dataset. Subsequently, we embarked on a fine-tuning process by adding additional layers to the base model. Diverse structures for the fine-tuning layers were experimented with, varying their type, position, and arguments to identify the optimal configuration. We explored several layer types and arrangements, with the most commonly used being 2D convolution, Global Average Pooling, Dropout, Batch Normalization, and others. The comprehensive architecture of the resulting model is illustrated in Figure 2.

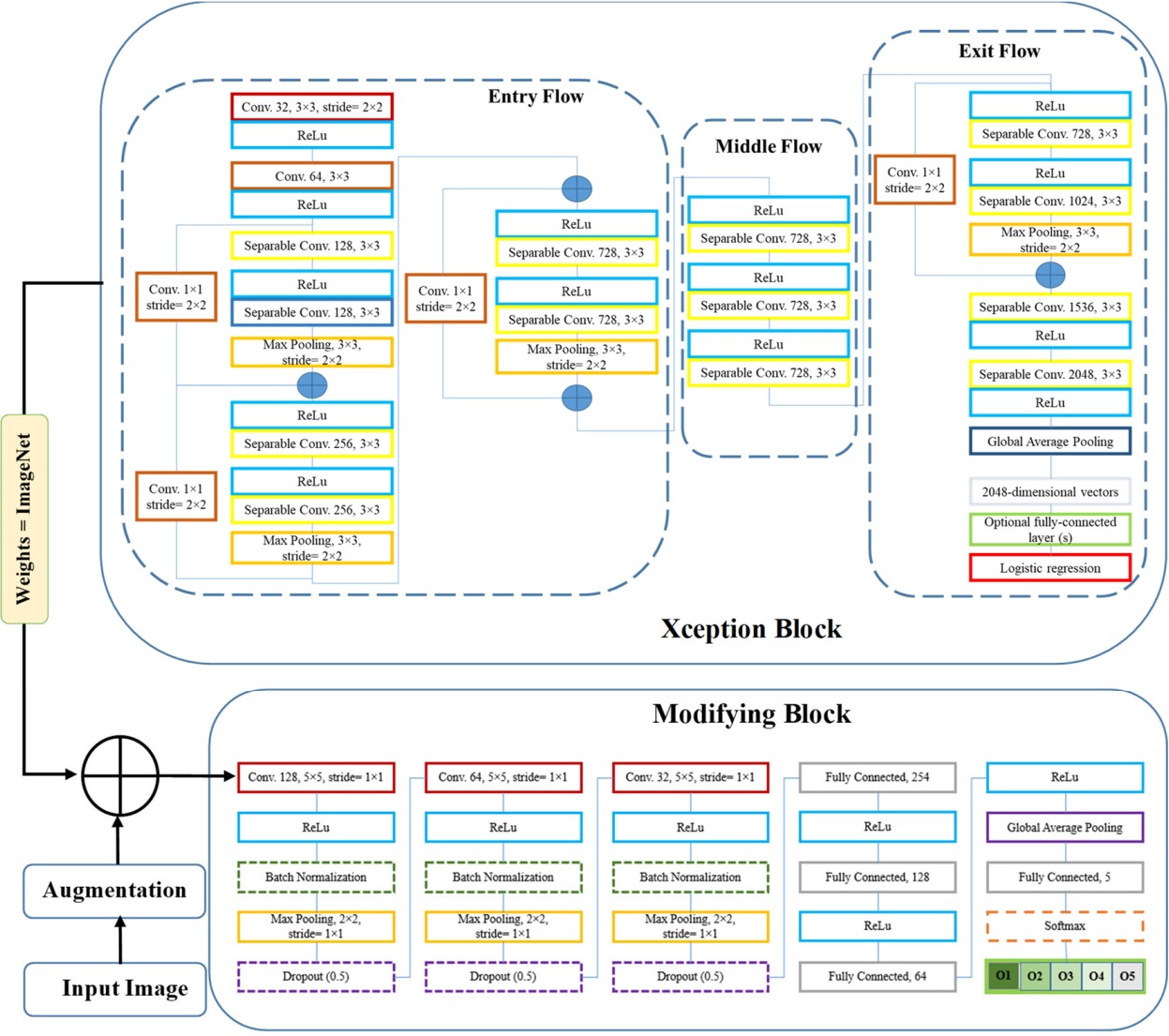

**Figure 2.** The architecture of the modified Xception deep learning model proposed for the study (Shapes with similar outlines correspond to layers with similar properties).

## 2.2. Xception Architecture

In this study, we employed the Xception deep learning architecture, a novel deep convolutional neural network model introduced by Google, Inc. (Mountain View, CA, USA) [28]. Xception features Depth Wise Separable Convolutions (DSC) to enhance performance and efficiency. Unlike traditional convolution, DSC divides the computation into two stages: depth wise convolution applies a single convolutional filter per input channel, followed by point wise convolution to create a linear combination of the depth wise convolution outputs.

Xception is a variant of the Inception architecture where Inception modules act as an intermediate step between regular convolution and DSC. With the same number of parameters, Xception surpasses Inception V3 on the ImageNet dataset due to its more efficient use of model parameters. The Xception architecture consists of 36 convolutional layers forming the feature extraction base of the network. In image classification tasks, the convolutional base is succeeded by a logistic regression layer. The 36 convolutional layers are organized into 14 modules, all with linear residual connections around them, excluding the first and last modules. In summary, the Xception architecture is a linear stack of depth wise separable convolution layers with residual connections [28].

## 2.3. Fine-Tuning and Modification

We first pre-trained the base model (Xception) using ImageNet weights. Next, the trainable attribute of all layers in the base model were frozen, ensuring that their weights remained fixed during training. This allowed us to use the pre-trained model as a starting point for further training on a new dataset. We then unfroze the last 20 layers in the Middle Flow and Exit Flow, making them trainable. By doing so, the pre-trained layers were prevented from overfitting on the new dataset while allowing the newly added layers to adapt to the new data. Finally, we added three blocks on top of the pre-trained base model, each containing Convolution, Batch Normalization, Max Pooling, and Dropout layers, followed by Fully Connected and Global Average Pooling layers. We called it the modifying block (Figure 2).

Table 3 provides detailed information about the various layers used, their output shapes, and the total number of parameters. The table covers both input image sizes studied (224 × 224 and 299 × 299). The developed model has approximately 27 million parameters for both image sizes, with only about 0.5% being non-trainable. Notably, Max Pooling, Dropout, and Global Average Pooling layers do not contribute to the total number of trainable parameters since they lack trainable parameters. As seen in Table 3, the number of parameters remains constant across both input image sizes, because the CNN architectures were totally the same for the two input image sizes.

**Table 3.** The detailed properties of the CNN architecture (Modified Xception) for two different input image sizes.

| Layer (Type) | Output Shape (Input Size = 224 × 224) | Output Shape (Input Size = 299 × 299) |
|---|---|---|
| Xception Block | (None, 7, 7, 2048) | (None, 10, 10, 2048) |
| Convolution 2D | (None, 7, 7, 128) | (None, 10, 10, 128) |
| Batch Normalization | (None, 7, 7, 128) | (None, 10, 10, 128) |
| Max Pooling 2D | (None, 4, 4, 128) | (None, 5, 5, 128) |
| Dropout | (None, 4, 4, 128) | (None, 5, 5, 128) |
| Convolution 2D | (None, 4, 4, 64) | (None, 5, 5, 64) |
| Batch Normalization | (None, 4, 4, 64) | (None, 5, 5, 64) |
| Max Pooling 2D | (None, 2, 2, 64) | (None, 3, 3, 64) |
| Dropout | (None, 2, 2, 64) | (None, 3, 3, 64) |
| Convolution 2D | (None, 2, 2, 32) | (None, 3, 3, 32) |
| Batch Normalization | (None, 2, 2, 32) | (None, 3, 3, 32) |
| Max Pooling 2D | (None, 1, 1, 32) | (None, 2, 2, 32) |
| Dropout | (None, 1, 1, 32) | (None, 2, 2, 32) |

**Table 3.** *Cont.*

| Layer (Type) | Output Shape (Input Size = 224 × 224) | Output Shape (Input Size = 299 × 299) |
|---|---|---|
| Dense | (None, 1, 1, 254) | (None, 2, 2, 254) |
| Dense | (None, 1, 1, 128) | (None, 2, 2, 128) |
| Dense | (None, 1, 1, 64) | (None, 2, 2, 64) |
| Global Average Pooling 2D | (None, 64) | (None, 64) |
| Dense | (None, 5) | (None, 5) |
| Total Parameters: | 27,721,803 | |
| Trainable Parameters: | 27,577,643 | |
| Non-trainable Parameters: | 144,160 | |

*2.4. Network Training*

To optimize the performance of the deep learning model for classifying olive fruits based on their ripening stages, several aspects required careful consideration. First, we needed to select the most appropriate optimizer among popular choices such as RMSprop, SGD, Adam, and Nadam. Accuracy was chosen as the evaluation metric to assess the model's performance. Additionally, we employed the categorical cross-entropy function as the loss function.

Training the model involved a series of experiments to identify the best combination of hyper parameters and architectural components. Initially, we trained the base model (Xception) using a batch size of 8 and 20 epochs. Subsequently, we trained the modified model with a batch size of 32 and 80 epochs (with an optional extension to 100 epochs for Model 1). Throughout the training process, we monitored the loss and accuracy trends for both the train and validation datasets at each epoch. This allowed us to analyze the models' performance and make informed decisions regarding their suitability for our task. Four promising candidates emerged from our experiments, each distinguished by its unique combination of image size and optimizer. They were:

- Model 1: Best performer with 224 × 224 image size and Nadam optimizer
- Model 2: Best performer with 224 × 224 image size and SGD optimizer
- Model 3: Best performer with 299 × 299 image size and RMSprop optimizer
- Model 4: Best performer with 299 × 299 image size and SGD optimizer

When evaluating these models, we considered multiple factors, such as accuracy, loss, and resistance to overfitting. Accuracy measures the proportion of correctly predicted instances, while loss represents the average error per instance. A lower loss value generally indicates better model performance. However, a model with high accuracy but relatively high loss may still encounter challenges in unseen data, signaling potential overfitting issues. Therefore, we assessed the risk of overfitting when comparing the four candidates.

The training, development, and testing procedures were executed using Python 3.7.10 and the Google Colab environment (K80 GPU and 12 GB RAM) with Keras, TensorFlow backend (version 2.13.0), OpenCV, and other relevant libraries.

**3. Results and Discussion**

This section presents the methodological approach taken to develop and evaluate deep learning models for classifying olive fruits according to their ripeness levels. Our next step is analyzing the results and discussing the implications of our findings.

*3.1. Training Progress*

The trend of losses and accuracies against the number of epochs for both the training and validation datasets and for the four candidate models are illustrated in Figures 3 and 4. For each candidate model, the minimum losses and maximum accuracies for training and validation data and the corresponding epochs are mentioned in Table 4.

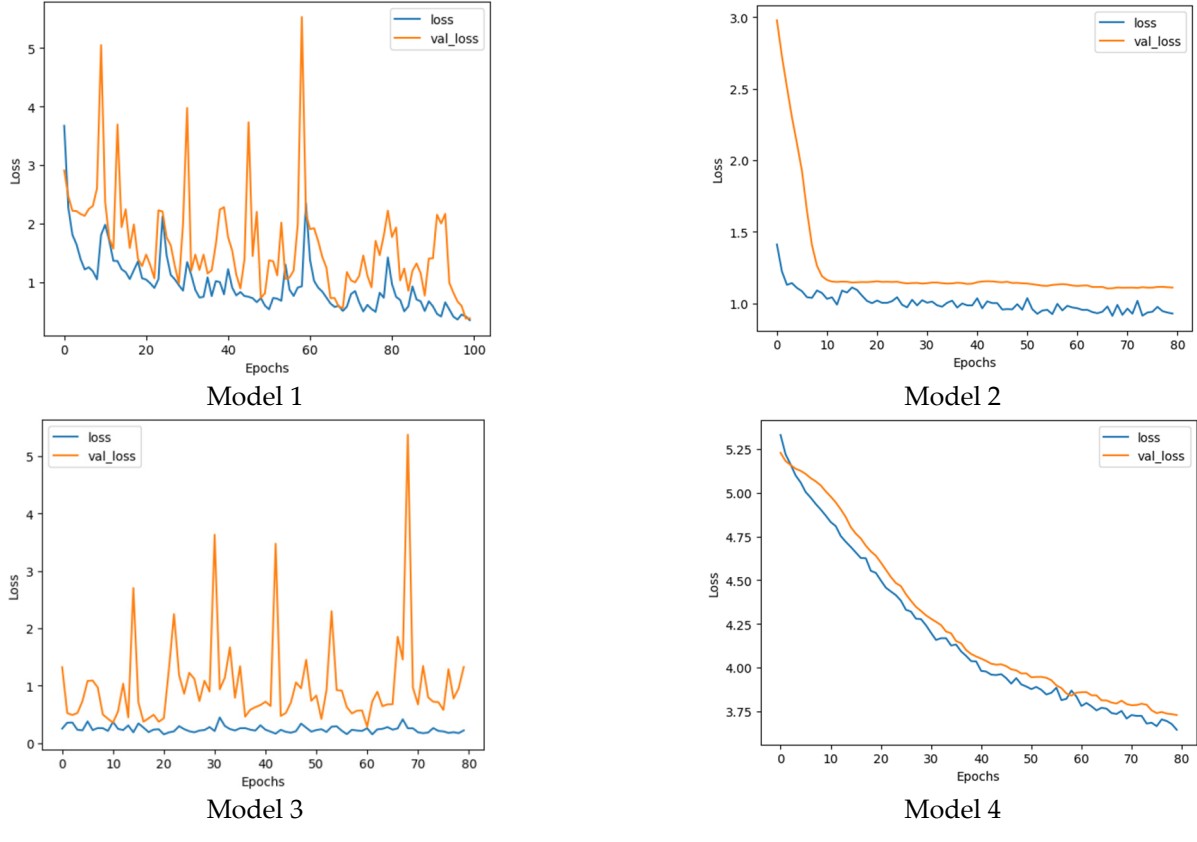

**Figure 3.** Loss trends for the four final candidate models.

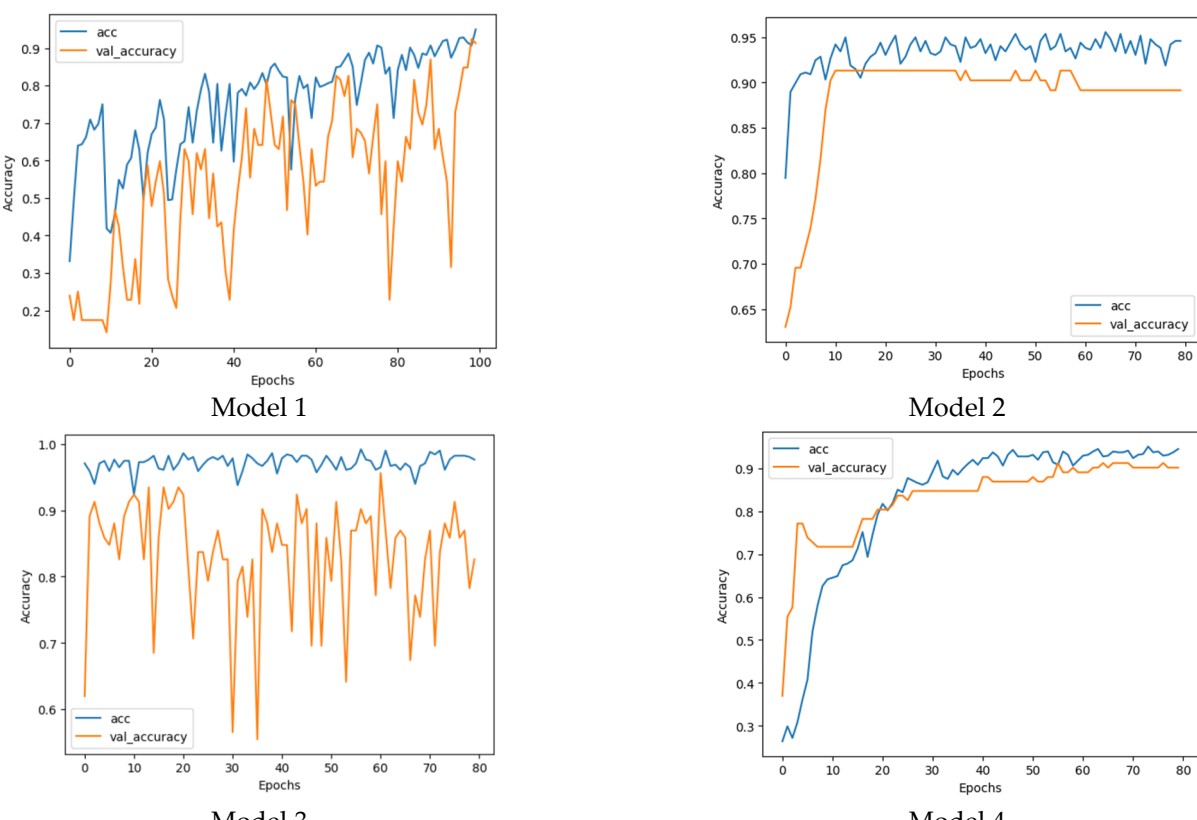

**Figure 4.** Accuracy trends for the four final candidate models.

**Table 4.** The values and epochs of minimum training and validation losses, as well as maximum training and validation accuracies for each candidate model.

|         | Min Train Loss/Epoch | Min Validation Loss/Epoch | Max Train Accuracy/Epoch | Max Validation Accuracy/Epoch |
|---------|----------------------|---------------------------|--------------------------|-------------------------------|
| Model 1 | 0.31/99              | 0.32/99                   | 0.95/97                  | 0.88/96                       |
| Model 2 | 0.42/73              | 1.23/66                   | 0.94/63                  | 0.92/56                       |
| Model 3 | 0.15/56              | 0.35/60                   | 0.99/55                  | 0.95/60                       |
| Model 4 | 1.61/79              | 3.60/79                   | 0.94/74                  | 0.91/75                       |

According to Figures 3 and 4, Models 1 and 3 exhibit substantial fluctuations in validation losses and validation accuracies, whereas Models 2 and 4 display a consistent downward trend in losses and a steady increase in accuracies, with only minor variations. These patterns suggest that Models 1 and 3 are susceptible to overfitting, whereas Models 2 and 4 are more resistant to it, making them more generalized and reliable in handling new, unseen data.

*3.2. Comparison of the Candidate Models*

Four candidate models perform differently on unseen (test) data. The result of loss and accuracy values on test data are provided in Table 5.

**Table 5.** Comparison of four final candidate models on test data.

|         | Test Loss | Test Accuracy |
|---------|-----------|---------------|
| Model 1 | 0.3938    | 0.9346        |
| Model 2 | 1.2338    | 0.8693        |
| Model 3 | 0.5502    | 0.9085        |
| Model 4 | 3.8232    | 0.8693        |

According to Table 5, both image sizes can provide low losses and high accuracies. Model 1 has the lowest test loss (0.3938) and highest test accuracy (0.9346), indicating good performance on the test set. However, test loss and test accuracy should not be the only factors considered when evaluating a CNN model, as another important factor is the risk of overfitting, which affects the model's generalization ability to new, unseen data. Therefore, the best model should be chosen based on a trade-off between test loss, test accuracy, and the possibility of overfitting. Model 1 cannot be the final choice since it possesses the possibility of overfitting during the training process, which may result in poor generalization to new data (Figures 4 and 5). Model 2 has a higher test loss (1.2338) and lower test accuracy (0.8693) compared to Model 1, but it does not show any signs of overfitting (Figures 4 and 5), indicating better generalization to new data. Model 3 has a slightly higher test loss (0.5502) and lower test accuracy (0.9085) than Model 1, but like Model 1, it displays the possibility of overfitting during training. Model 4 and Model 2 do not have an overfitting issue. Model 4 has a significantly higher test loss (3.8232) but a comparable value of test accuracy, indicating poor performance on the test set.

To evaluate the performance of the candidate models in discriminating between the different classes (O1 to O5), we utilized four parameters: true positives (TP), true negatives (TN), false positives (FP), and false negatives (FN). These parameters are used to calculate two classification metrics: classification report and confusion matrix. The classification report provides information about the performance of a model through precision, recall, and F1-score, as described by Equations (1)–(3). Precision measures how well the model predicts positive cases, while recall measures the proportion of correctly predicted positive instances out of the total actual positive instances. The F1-score is the harmonic mean of precision and recall, providing a balanced measure that combines both metrics.

$$Precision = \frac{TP}{TP + FP} \, . \tag{1}$$

$$Recall = \frac{TP}{TP + FN}. \tag{2}$$

$$F1 - score = 2 \times \frac{Precision \times Recall}{Precision + Recall}. \tag{3}$$

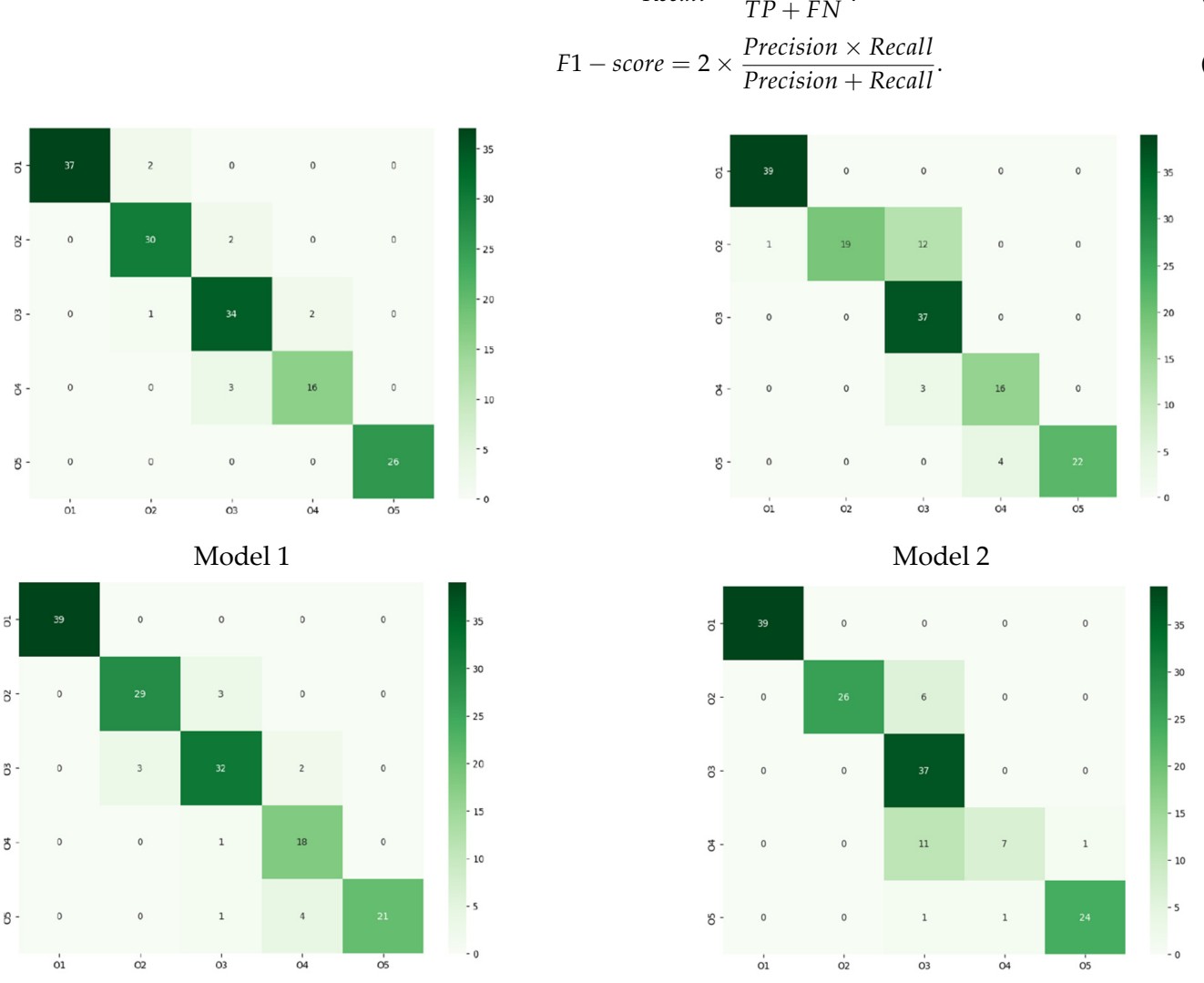

**Figure 5.** Confusion matrix for the four final candidate models.

The classification report for recognizing the five olive classes under study is presented in Table 6. According to this table, Model 1, 3, and 4 achieved a precision value of 1.00 for the O1 class, while Model 2 had a slightly lower precision of 0.97. This indicates that all models performed well in predicting the O1 class. In recognizing class O2, Models 2 and 4 achieved a perfect accuracy of 100%, while Models 1 and 3 had an accuracy of 91%. Class O3 was better identified by Models 1 and 3, with an accuracy of 89% for both models, whereas Models 2 and 4 had a lower accuracy of 73% and 69%, respectively. For class O4, all models performed similarly, with an accuracy ranging from 87% to 93%. Finally, class O5 was identified perfectly by all models, except for Model 4, which had an accuracy of 96%. The performance of all models in identifying classes O1 and O5 was nearly perfect, likely due to their distinct visual properties.

To assess the models' ability to avoid false negatives, we can compare their recall values. Models 2, 3, and 4 correctly predicted all O1 instances as O1, meaning they had zero false negatives. Model 1 had a recall of at least 90% for class O1. For class O2, Model 2 had a lower recall of 59%, while Models 1, 3, and 4 achieved recalls of 0.94, 0.91, and 0.81, respectively. Models 2 and 4 were perfect in predicting class O3, while Models 1 and 3 had a high accuracy. In case of class O4, Model 4 performed weakly with a recall of 37%, while the other models had a reasonable performance. Finally, all O5 instances were correctly

predicted as O5 by Model 1, and Models 2 and 4 were very good, while Model 3 had a relatively lower accuracy of 77%. Overall, the results suggest that all models performed well in recognizing classes O1 and O5, while there were variations in performance across the other classes.

**Table 6.** Classification report for recognizing the classes by the four final candidates.

| | Precision | | | | Recall | | | | F1-Score | | | | Support |
|---|---|---|---|---|---|---|---|---|---|---|---|---|---|
| | Model 1 | Model 2 | Model 3 | Model 4 | Model 1 | Model 2 | Model 3 | Model 4 | Model 1 | Model 2 | Model 3 | Model 4 | All |
| O1 | 1.00 | 0.97 | 1.00 | 1.00 | 0.90 | 1.00 | 1.00 | 1.00 | 0.95 | 0.99 | 1.00 | 1.00 | 39 |
| O2 | 0.91 | 1.00 | 0.91 | 1.00 | 0.94 | 0.59 | 0.91 | 0.81 | 0.92 | 0.75 | 0.91 | 0.90 | 32 |
| O3 | 0.89 | 0.73 | 0.89 | 0.69 | 0.92 | 1.00 | 0.86 | 1.00 | 0.91 | 0.84 | 0.88 | 0.81 | 37 |
| O4 | 0.89 | 0.80 | 0.82 | 0.88 | 0.84 | 0.84 | 0.95 | 0.37 | 0.86 | 0.82 | 0.88 | 0.52 | 19 |
| O5 | 1.00 | 1.00 | 1.00 | 0.96 | 1.00 | 0.85 | 0.77 | 0.92 | 1.00 | 0.92 | 0.87 | 0.94 | 26 |
| Micro Avg. | 0.94 | 0.88 | 0.93 | 0.88 | 0.92 | 0.87 | 0.90 | 0.87 | 0.93 | 0.87 | 0.91 | 0.87 | 153 |
| Macro Avg. | 0.94 | 0.90 | 0.92 | 0.90 | 0.92 | 0.86 | 0.90 | 0.82 | 0.93 | 0.86 | 0.91 | 0.83 | 153 |
| Weighted Avg. | 0.94 | 0.90 | 0.93 | 0.90 | 0.92 | 0.87 | 0.90 | 0.87 | 0.93 | 0.87 | 0.91 | 0.86 | 153 |
| Samples Avg. | 0.92 | 0.87 | 0.90 | 0.87 | 0.92 | 0.87 | 0.90 | 0.87 | 0.92 | 0.87 | 0.90 | 0.87 | 153 |

In summary, Model 2 appears to be the most suitable choice among the four models, as it demonstrates low test loss, high test accuracy, and no signs of overfitting. Additionally, it achieves a relatively high F1-score, indicating its ability to accurately classify instances across all classes.

Figure 5 displays the confusion matrices for the classification of five olive classes using four candidate models. The columns represent the predicted values, while the rows show the true values. Upon examining the matrices, we observed that all models achieved a perfect classification (100%) for O1, with the exception of Model 1, which mislabeled two instances as O2. Moving on to class O2, all models primarily confused it with O3. Model 1 had the least confusion, while Model 2 had the most. For O3, Models 2 and 4 successfully recognized it, but Models 1 and 3 mixed up a few instances with O2 and O4. In relation to class 4, Models 1 and 2 performed similarly, each with only three instances confused with O3. Model 4 showed the weakest performance in classifying O4, while Model 3 performed the best, mistakenly classifying just one instance as O3. Lastly, in the case of O5, Model 1 achieved the highest accuracy with 100%, but Model 2 misidentified four O5 instances as O4. Model 3 confused one O5 instance with O3 and four with O4, while Model 4 misclassified one instance with both O3 and O4. To summarize, the models generally demonstrated good recognition for classes O1 and O5, which can be attributed to the minimal visual similarities compared to the other classes.

Our proposed network for detecting olive fruits ripening stage demonstrates competitive performance metrics, exhibiting similar or enhanced accuracy or precision compared to prior studies such as Guzmán et al. [9] in predicting olive maturity using specific algorithms, and Puerto et al. [20] for classifying Veraison and visible defects of olive fruit under an online system. The robustness of the Xception deep learning method in image-based classification tasks has been demonstrated in various applications. For instance, Pujari et al. [29] achieved 99.01% accuracy in classifying breast histopathological images using this method. Similarly, Wu et al. [14] applied the algorithm to classify scene images with an accuracy of 92.32%, and Salim et al. [27] utilized the Xception model to classify fruit species with a total accuracy of 97.73%. A comparative analysis, including the most relevant works on the olive, is provided in Table 7, which discusses classification tasks using both traditional and modern image analysis methods. One significant advantage of deep learning-based image processing techniques is their independence from environmental conditions such as lighting, background, distance to object, camera properties, etc., unlike traditional image processing methods. Additionally, deep learning techniques excel in feature extraction, allowing them to extract high-level features compared to low-level features like edges and color extracted by traditional methods. This high-level feature extraction makes deep learning techniques powerful and robust tools for processing images captured in unstructured and natural conditions, enabling their application in more challenging scenarios. Guzmán

et al. [9] used traditional image processing techniques to extract maturity indexes of olive fruit but were unable to extract high-level features, making it unsuitable for maturity-based classification tasks. Puerto et al. [23] applied traditional image processing techniques for classifying different batches of olives entering the milling process but could not identify the individual olive pieces within the same batch. The most relevant work in this context was carried out by Khosravi et al. [17] who successfully classified on-branch olives based on maturity stage (91.91%), although the method's robustness for post-harvest sorting remains unclear. Our proposed network, a modified version of Xception, demonstrates promising performance metrics in recognizing olive fruit ripening stages based on color images. This indicates its reliability as a tool for post-harvest sorting of olive fruits.

**Table 7.** Comparative analysis of image-based classification technologies in olives.

| | Classification Target | Classification Technology | Structured Condition | Extractable Features Level |
|---|---|---|---|---|
| Guzmán et al. [9] | Maturity index of olives | Traditional image processing | Yes | Low |
| Puerto et al. [23] | Olive species | Traditional image processing + ANN | Yes | Low |
| Khosravi et al. [17] | On-branch olive fruit maturity stage | Deep learning (lightweight CNN) | No | High |
| Our work | Post-harvest olive fruit maturity stage | Deep learning (Xception) | No | High |

## 4. Conclusions

Olives are a vital crop with various post-harvest applications, including pickling, canning, and oil production, each requiring a specific ripening stage. To address this challenge, a reliable classification system is crucial to sort olives according to their maturity levels. This study aimed to develop an automated deep learning model utilizing color images to classify 'Roghani', an Iranian olive cultivar, into five ripening stages. We employed a modified and fine-tuned Xception architecture, harnessing cutting-edge image processing and deep learning techniques to effectively categorize olives. Four Xception-based models were shortlisted and evaluated based on their performance, using metrics such as loss, accuracy, classification reports, confusion matrices, and overfitting risk. While all four models showed comparable performance, Model 1 stood out. However, considering model generality and stability, Model 1 raised concerns due to substantial fluctuations in validation losses and accuracies during training, indicating a high risk of overfitting. Model 3 boasted a remarkable accuracy, but its reliability was compromised by its susceptibility to overfitting. Models 2 and 4 demonstrated stable validation losses and accuracies throughout training, rendering them superior in terms of generality and stability. Although their accuracies were not the highest among all models, they were still satisfactory. Of the two, Model 2 is preferred owing to its lower loss value. When selecting a model, a trade-off between classification performance and model generality must be considered. For the present study, Model 2 emerges as the optimal choice, striking a balance between respectable classification results and minimal risk of overfitting, suggesting that it may generalize well to unseen data. The findings of this research constitute a significant breakthrough in olive sorting and classification, providing a potent tool for enhancing the efficiency and precision of olive processing and production.

**Author Contributions:** Conceptualization, S.I.S. and M.R.; data curation, M.R.; formal analysis and modelling, S.I.S.; writing—original draft preparation, S.I.S. and M.R.; writing—review and editing, S.I.S. and M.R. All authors have read and agreed to the published version of the manuscript.

**Funding:** No funding was received for this work.

**Institutional Review Board Statement:** Not applicable.

**Data Availability Statement:** The data supporting this study is available upon request.

**Conflicts of Interest:** There are no conflicts of interest to declare.

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
