# Peer review of "A Modified Xception Deep Learning Model for Automatic Sorting of Olives Based on Ripening Stages"

_inventions, doi:10.3390/inventions9010006_

Round 1

Reviewer 1 Report

Comments and Suggestions for Authors

Author Response

Hello,

Special thanks to you and the reviewers for valuable comments. In the following, we respond to the comments of the reviewers in order (blue texts). All changes to the revised manuscript have been yellow-highlighted.

   Reviewer #1:

Xception deep learning model for automatic sorting of olives based on ripening stages

This work proposed Xception deep learning architecture for predicting the five stages of olives. The motivation and background for addressing the problem and providing a solution as a computer vision-based system, are relevant and strong. However, I am skeptical about the author’s viewpoint on finetuning and overfitting.

Following are the issues found in the manuscript in chronological order:

Introduction

1- The motivation for the problem of olive stage sorting is very clearly addressed, and the need to develop an image vision system for sorting olives based on ripeness stages is presented substantially. However, it would be nice to summarize the limitations of the existing studies in the literature.

Response:

A description about the limitations of the existing studies has been added in the literature.Page 3

2- Authors should check the section for typographical errors such as the last line on page 2, “Veraison”, second line on page 3, “75%”.

Response:

It was checked and corrected.

3- The literature presented in the manuscript points to the fact that ANN has been predominantly used, with only one study using Inception-Resnet architecture for solving this problem. Though the authors did mention how Xception architecture is different and better than the Inception network, the reason for not using other computer vision deep learning architectures was not presented.

So, how does the author reach the conclusion of using the Xception architecture? Is it empirical?

Response:

To find the best model, multiple standard structures were experimented. But the results of the discarded standard models were not mentioned for brevity. However, It has been pointed out in the paper that the Xception is a powerful neural network that excels in image classification tasks owing to its efficiency and accuracy (Chollet, 2017).
Materials and Methods

Section 2.1

4- It was mentioned that data augmentation strategies were applied, but the size of the data was not specified after augmentation. Please update the manuscript with the information about the increased size of data after augmentation.

Response:

The resultant augmented images were not saved in memory, and the code used for data augmentation saved the data in virtual memory. So, the accurate number of augmented images is not available. It is worth mentioning that data augmentation should be exclusively applied to the training dataset to increase its size and diversity, while the validation dataset should remain unaltered to accurately assess the model's performance on unseen data.

5- The authors should check the percentages of training, validation, and test datasets since the percentages mentioned in the manuscript do not sum up to 100%, but rather 95%.

Response:

According to the data splitting method, 20% of the total number of input images (761) i.e. 153 images were considered as the test dataset. The remaining data were 609 images, from which 20% i.e. 92 images were considered as validation set, and the remaining images i.e. 516 images were considered for training set.

6- It was stated that “We normalized the images by dividing their size by the maximum size of captured images”. Usually, the pixel values are normalized. Authors should check this statement,
Response:

Thanks! the manuscript was revised for this issue.

7- In Table 2, please check the value of Random rotation; it must be an angle, not a value. If 0.15 is the angle, it does not make sense, as it will not change the image orientation appropriately.

Response:

The term "0.15 for random rotation" refers to the probability of applying a random rotation transformation to the image during the augmentation process. The probability is set to 0.15, which means that there is a 15% chance that a random rotation transformation will be applied to the image.

Section 2.2

8- The authors have provided an explanation of freezing the weights of the base network, but why the layers corresponding to the Middle and Exit flow were trained, since it is also part of the Xception network?

Response:

The trainable attribute of all layers in the base model were frozen to ensure that their weights remained fixed during training. This allowed us to use the pre-trained model as a starting point for further training on a new dataset. We then unfroze the last 20 layers in the Middle Flow and Exit Flow, making them trainable. By doing so, the pre-trained layers were prevented from overfitting on the new dataset while allowing the newly added layers to adapt to the new data.

9- Usually, the transfer learning approach changes the last layer of the architecture, but in this case, the authors have added a CNN architecture. Is there any specific reason for this?

Response:

The process of adding a CNN architecture was among the main novelties of this study, as it aims at modifying the base architecture. By doing so, the model performed well and provide high accuracies.
10- The deep learning architecture is proposed for the two different image inputs in terms of dimensions, still, the number of the network’s parameters is the same despite different CNN architectures cascaded by the Xception network. This needs to be explained in detail. How?

Response:

The number of parameters remains constant across both input image sizes, because the CNN architectures cascaded by the Xception network were totally the same for the two input image sizes.

Section 2.3

11- There is something wrong with the following statement: “We trained the model without fine-tuning using a batch size of 8 and 20 epochs.” I am skeptical about this statement. What do the authors want to convey here? When a pre-trained network is trained with any data, it is called fine-tuning. Even if the entry flow module weights of the Xception network are frozen and its middle and exit flow are trained, it will still be called fine-tuning.

Response:

You are right! We used the concept of fine-tuning also for the process of adding layers to the end of the base model. We should use the term “extending or modifying the architecture of the model” rather than “fine-tuning”. The manuscript was revised to take into account this problem.

Results

Section 3.1

12- The following statement also looks doubtful: “A model with high accuracy but excessively high loss may still encounter challenges in unseen data”. How is it possible to achieve high loss and accuracy both at the same time? The authors need to provide further explanations about this.

Response:

It is just a comparison between two scenarios: 1- high accuracy and ‘A’ loss, 2- high accuracy and ‘B’ loss, where A>B. We revised the sentence for more clarity.

13- The authors relate accuracy and loss fluctuations with overfitting. To the best of my knowledge, the stability of the method can be explained by fluctuations in accuracy and loss. High fluctuations mean a higher standard deviation. The higher standard deviation in results points to an unstable method. Overfitting can be defined by low loss and high accuracy during training and vice versa during validation or testing. How do fluctuations in accuracy and loss explain the overfitting? The authors need to provide further explanations about this.

Response:

Fluctuations in accuracy and loss during training are indeed among the overfitting symptoms in deep learning models. They are also symptoms of network instability. Monitoring these fluctuations and taking appropriate measures, such as regularization or weight decay, can help to mitigate overfitting and improve the model's generalization capabilities.

Section 3.2

14- How do the loss and accuracy of the test dataset define overfitting without considering the loss or accuracy of the training dataset?

Response:

Yes, the loss and accuracy of the test dataset do not define overfitting without considering the loss or accuracy of the training dataset. The manuscript is double-checked to take into account this inaccurate statement.

15- The authors used the term “perfect recall” on page 10. What does that mean?

Response:

It means recall value of 100%.

16- Are there any specific explanations about the lower recall and F1-score value of model 4 for the O4 stage?

Response:

Low recall and F1-score for O4 shows that the model performed weak in correctly identifying this class from the total number of actual positive samples in a dataset. This can be attributed to the fact that this class contains samples which have visual similarities with both O3 and O5 classes. Also, this can be due to similar growth attributes with class O3 and O5.

17- The authors have mentioned that model 2 did not show any signs of overfitting. What does that mean? What metrics did the authors use to reach this conclusion? It is necessary to be presented in the manuscript.

Response:

The most important factor for measuring overfitting was the trends of loss and accuracy in train phase. For model 2, these trends were based on the principal of neural networks, i.e. the loss experienced an almost steady decrease while the accuracy experienced an almost steady increase during the train phase. Additionally, minimum fluctuations were witnessed in these two trends.

18- The proposed network must be compared with existing relevant studies in literature.

Response:

This point has been considered in the revised manuscript. Page 13.

Conclusion

19- The findings presented in the conclusion are conceptually incorrect due to doubtful explanations about loss and accuracy and eventually overfitting.

Response:

Detailed description about this issue has been provided in the above. However, the conclusion part has been revised for taking into account this issue.

Reviewer 2 Report

Comments and Suggestions for Authors

This is an interesting and well written paper. The methodology sounds.
I have following remarks authors should address and discuss in the paper.

1. The process of maturing of olive fruit seems to be rather continues than stage-based. Did you consider modeling the process as a regression rather a classification? For example network response 0 might be a "young" (green) fruit while 1 is a "mature" (black) fruit. The between stages are inside [0,1] range.
2. Table 1 - please add information about the variance of average mass.
3. Please add implementation details and detailed setting of training and validation.
4. Please add more details about data acquisition. What are differences in lighting condition during acquiring images? Were all pictures taken on the blue background? Please discuss if and how it might affect the final accuracy of the solution and if the network will work when the environment where the pictures are taken slightly changes? Have you considered image augmentation during training?
5. Please make the experiment reproducible: publish both dataset and source codes in online repository

Author Response

Hello,

Special thanks to you and the reviewers for valuable comments. In the following, we respond to the comments of the reviewers in order (blue texts). All changes to the revised manuscript have been yellow-highlighted.

Reviewer #2:

This is an interesting and well written paper. The methodology sounds.
I have following remarks authors should address and discuss in the paper.
1- The process of maturing of olive fruit seems to be rather continues than stage-based. Did you consider modeling the process as a regression rather a classification? For example, network response 0 might be a "young" (green) fruit while 1 is a "mature" (black) fruit. The between stages are inside [0,1] range..                          

Response:

Our current approach employs a classification model to categorize olives into discrete stages based on identifiable characteristics, aligning with previous models in the literature on olive ripening. While olive maturation is a continuous process, utilizing a regression-based modeling approach within the [0,1] range could offer a more intricate depiction of the transitions between different ripeness levels. However, our preference for a classification model was primarily guided by the need to offer clear and distinct classifications, essential for practical applications in olive processing and sorting. The discrete stage classification facilitates straightforward decision-making, crucial for identifying specific ripeness stages vital in olive processing. Nonetheless, we recognize the potential advantages of a regression-based model in capturing the continuous nature of the maturation process. We see this as a promising avenue for future research to delve deeper into the nuanced spectrum of olive fruit ripening stages.

2- Table 1 - please add information about the variance of average mass.

Response:

This parameter is added in the revised manuscript.

3- Please add implementation details and detailed setting of training and validation.

Response:

20% of the total 761 input images i.e. 153 images were considered as the test dataset. The remaining data were 609 images, from which 20% i.e. 92 images were considered as validation set, and the remaining images i.e. 516 images were considered for training set. The training process involved passing the input data through several layers, obtaining the output, and comparing it with the desired output. The difference between the two, which served as the error, was then calculated. Using this error, the network parameters were adjusted and fed the data back into the network to compute new results and errors. This process was repeated multiple times, adjusting the parameters after each iteration to minimize the error. There are various formulas and functions to calculate the network error. Once the error was computed, the parameters were updated to move closer to minimizing it, that is, optimizing the weights to achieve the lowest possible error.

4- Please add more details about data acquisition. What are differences in lighting condition during acquiring images? Were all pictures taken on the blue background? Please discuss if and how it might affect the final accuracy of the solution and if the network will work when the environment where the pictures are taken slightly changes? Have you considered image augmentation during training?

Response:

One of the most important advantages of deep learning – based image processing techniques and convolutional neural networks is that unlike the traditional image processing methods, they are not dependent on environmental conditions including lighting, background, distance to object, camera properties, etc., making them more powerful and robust tools in processing images taken in natural condition. By addressing this fact, we did not consider a special capturing condition or structure. According to the text, we considered image augmentation to artificially increase the number of input images. This note has been added to the paper.

5- Please make the experiment reproducible: publish both dataset and source codes in online repository.

Response:

The dataset and source codes are available upon request.

Thank you for your consideration.

Sincerely

Round 2

Reviewer 1 Report

Comments and Suggestions for Authors

1. I would again ask authors to recheck the percentage of papers in agriculture, which was mentioned as 745%.

2. Although author explained about the number of images in train (20%), test (20%) and validation (20%) dataset, in the manuscript, the validation data percentage is written as 15%. Please check.

3.Please check if authors have used tf.keras.layers.RandomRotation or tf.keras.preprocessing.image.random_rotation, both are different in terms of applicability. It seems the authors have used former one, which applies rotation to feature maps in the layers, not the images.

4. Although the number of parameters in Xception network are the same, but number of parameters in the proposed cascaded CNN architecture are different. So summing up all parameters (Xception+cascaded CNN architecture) should produce different number of parameters.

5. The perfect recall is achieved by model 2, 3, and 4, not model 1. Please check the related text.

6. The authors have mentioned in the response letter that the proposed architecture was compared with relevant methods in the architecture. The authors might have forgotten to add the results. Therefore, authors should include the comparative analysis, and discuss the results.

Author Response

Thanks for valuable comments. All changes to the revised manuscript have been yellow-highlighted.

  1. I would again ask authors to recheck the percentage of papers in agriculture, which was mentioned as 745%.

We check it again, the following is senntence of Benos, et al (2021). Machine Learning in Agriculture: A Comprehensive Updated Review. In Sensors (Vol. 21, Issue 11). https://doi.org/10.3390/s21113758) which used for that:

As will be discussed next, overall, a 745% increase in the number of journal papers took place in the last three years as compared to [12], thus justifying the need for a new updated review on the specific topic. Moreover, crop management remained as the most investigated topic, with a number of ML algorithms having been exploited as a means of tackling the heterogeneous data that originated from agricultural fields. As compared to [12], more crop and animal species have been investigated by using an extensive range of input parameters coming mainly from remote sensing, such as satellites and drones. In addition, people from different research fields have dealt with ML in agriculture, hence, contributing to the remarkable advancement in this field.

  1. Although author explained about the number of images in train (20%), test (20%) and validation (20%) dataset, in the manuscript, the validation data percentage is written as 15%. Please check.

Thank you for your consideration. The mistake has happened in the previous response, and the manuscript is correct. The following text has been added to the manuscript to provide a clearer explanation of the data splitting procedure:

The image data was divided into three distinct parts: the training set, the validation set, and the testing set. To accomplish this, 20% of the total data (equivalent to 153 images) was assigned to the test dataset. The remaining data consisted of 609 images, with 15% (92 images) being allocated for the validation set, and the remaining 516 images being utilized for the training set.

3.Please check if authors have used tf.keras.layers.RandomRotation or tf.keras.preprocessing.image.random_rotation, both are different in terms of applicability. It seems the authors have used former one, which applies rotation to feature maps in the layers, not the images.

Yes, we used the following code for data augmentation that includes all arguments:

img_augmentation = Sequential(

    [        layers.RandomRotation(factor=0.15),

        layers.RandomTranslation(height_factor=0.1, width_factor=0.1),

        layers.RandomFlip(),

        layers.RandomContrast(factor=0.15),    ],

    name="img_augmentation",)

  1.  Although the number of parameters in Xception network are the same, but number of parameters in the proposed cascaded CNN architecture are different. So summing up all parameters (Xception+cascaded CNN architecture) should produce different number of parameters.

In the manuscript, we have claimed that the number of parameters remains constant across both input image sizes, because the CNN architectures (modified Xception) were totally the same for the two input image sizes. The number of parameters in a CNN model remains constant across different input image sizes because the parameters are shared and reused throughout the network. In a CNN, the parameters (weights and biases) are applied to each region (or kernel) of the input image through convolutional layers. These kernels slide over the entire image, extracting features at each location. The parameters are shared across all locations, meaning the same weights and biases are used for each kernel at every position. When the input image size changes, the number of regions the kernels slide over may vary, but the parameters themselves do not change. The weights and biases are learned during the training process and remain fixed regardless of the input image size. This allows the model to efficiently extract and generalize features from images of different sizes without needing to adjust the number of parameters.

To better convey all our novelties in this paper, we’d better change the title of the paper to this:

Modified Xception deep learning model for automatic sorting of olives based on ripening stages

  1. The perfect recall is achieved by model 2, 3, and 4, not model 1. Please check the related text.

Well, thank you. The text was revised.

  1. The authors have mentioned in the response letter that the proposed architecture was compared with relevant methods in the architecture. The authors might have forgotten to add the results. Therefore, authors should include the comparative analysis, and discuss the results.

We did add this in the manuscript but in a non-suitable place. So, we transferred this paragraph to the last paragraph just before the Conclusion section.

Submission Date

12 November 2023

Reviewer 2 Report

Comments and Suggestions for Authors

Authors have addressed my remarks. In my opinion paper can be accepted in the present form.

Author Response

Thanks

Round 3

Reviewer 1 Report

Comments and Suggestions for Authors

The authors have addressed all the comments. However, the manuscript still missing the comparative analysis with the state-of-the-art methods corresponding to the olives sorting problem. This is very important to prove the efficacy of the presented method. 

Author Response

Dear Reviewer,

Thank you for your time and constructive feedback. We have now incorporated a detailed comparative analysis into the revised manuscript, highlighted specifically between lines 366 and 395. Additionally, we have included a new table, Table 7, providing a comprehensive comparison with pertinent existing methods related to olives sorting and classification.

Best regards